# Global TALES feasibility study: Personal narratives in 10-year-old children around the world

**Marleen F. Westerveld**[1]*, **Rena Lyons**[2☯], **Nickola Wolf Nelson**[3☯], **Kai Mei Chen**[4], **Mary Claessen**[5], **Sara Ferman**[6], **Fernanda Dreux M. Fernandes**[7], **Gail T. Gillon**[8,9], **Khaloob Kawar**[10], **Jelena Kuvač Kraljević**[11], **Kakia Petinou**[12], **Eleni Theodorou**[12], **Tatiana Tumanova**[13], **Ioannis Vogandroukas**[14,15], **Carol Westby**[16☯], on behalf the Global TALES Consortium[¶]

**1** Griffith Institute for Educational Research, Griffith University, Gold Coast, Queensland, Australia, **2** Discipline of Speech and Language Therapy, School of Health Sciences, College of Medicine, Nursing and Health Sciences, NUI Galway, Galway, Ireland, **3** Department of Speech, Language, and Hearing Sciences (Emeritus), College of Health and Human Services, Western Michigan University, Kalamazoo, Michigan, United States of America, **4** Department of Speech Language Pathology and Audiology, Chung Shan Medical University, Taichung, Taiwan, R.O.C, **5** Curtin School of Allied Health, Curtin University, Perth, Western Australia, Australia, **6** Department of Communication Disorders, Sackler Faculty of Medicine, Tel-Aviv University, Tel-Aviv, Israel, **7** Department of Communication Sciences and Disorders, Occupational Therapy and Physical Therapy, School of Medicine, University of Sao Paulo, Sao Paulo, Brazil, **8** Child Well-being Research Institute, University of Canterbury, Christchurch, New Zealand, **9** Better Start National Science Challenge, Liggins Institute, University of Auckland, Auckland, New Zealand, **10** Education Department, Beit Berl College, Kfar Sava, Israel, **11** Department of Speech and Language Pathology, Faculty of Education and Rehabilitation Sciences, University of Zagreb, Zagreb, Croatia, **12** Department of Rehabilitation Sciences, School of Health Sciences, Cyprus University of Technology, Limassol, Cyprus, **13** Logopedics Department, Institute of Childhood, Moscow State University of Education, Moscow, Russia, **14** School of Education Nicosia University, Nicosia, Cyprus, **15** Department Health Care and Social Work New Bulgarian University, Sofia, Bulgaria, **16** Bilingual Multicultural Services, Albuquerque, New Mexico, United States of America

☯ These authors contributed equally to this work.
¶ Members of the Global TALES Consortium are listed in the Acknowledgments.
* m.westerveld@griffith.edu.au

**Data Availability Statement:** Some data (transcripts, question 4) cannot be shared publicly because of confidentiality and privacy concerns. The Griffith University Human Ethics Research

## Abstract

Personal narratives make up more than half of children's conversations. The ability to share personal narratives helps build and maintain friendships, promotes physical and emotional wellbeing, supports classroom participation, and underpins academic success and vocational outcomes. Although personal narratives are a universal discourse genre, cross-cultural and cross-linguistic research into children's ability to share personal narratives is in its infancy. The current study addresses this gap in the research by developing the Global TALES protocol, a protocol comprising six scripted prompts for eliciting personal narratives in school-age children (excited, worried, annoyed, proud, problem situation, something important). We evaluated its feasibility with 249 ten-year-old children from 10 different countries, speaking 8 different languages, and analyzed researchers' views on the process of adapting the protocol for use in their own country/language. At group-level, the protocol elicited discourse samples from all children, although individual variability was evident, with most children providing responses to all six prompts. When investigating the topics of

Committee approved the study, and the consent document that was approved assured participants that their data would not be shared with third parties beyond the research team. To request data underlying this manuscript, please contact either Dr. Marleen Westerveld, or the Manager, Research Ethics, Griffith University Human Research Ethics Committee at research-ethics@griffith.edu.au or +61-7-37354375, reference 2018/273.

**Funding:** MW has a financial relationship with SALT Software and used these funds to support this project. https://www.saltsoftware.com/ GG received funding from the Better Start National Science Challenge Ministry of Education Business Innovation and Employment (Grant Number 15-02688). The funders had no role in study design, data collection and analysis, decision to publish, or preparation of the manuscript. No other authors received specific funding for this work.

**Competing interests:** I declare that I have read the journal's policy and the authors of this manuscript have the following competing interests: MW declares that she has a consultancy agreement with SALT Software LLC for which she receives compensation. However, SALT Software provided no funding for the current project and had no direct influence on this work. Dr Westerveld's consulting agreement does not alter our adherence to PLOS ONE policies on sharing data and materials. The other authors declare no competing interests exist.

children's personal narratives in response to the prompts, we found that children from around the world share many commonalities regarding topics of conversation. Once again individual variability was high, indicating the protocol is effective in prompting children to share their past personal experiences without forcing them to focus on one particular topic. Feedback from the participating researchers on the use of the protocol in their own countries was generally positive, although several translation issues were noted. Based on our results, we now invite clinical researchers from around the world to join us in conducting further research into this important area of practice to obtain a better understanding of the development of personal narratives from children across different languages and cultures and to begin to establish local benchmarks of performance.

## Introduction

Personal narratives, defined as accounts of personally experienced events, are one of the most spontaneous and earliest developing forms of discourse [1], making up more than half of children's conversations [2]. Personal narratives assist people in understanding and processing experiences [3]. As Fivush et al. [4] explained "it is as we create organised, explanatory accounts of actions in the world, which are integrated with subjective thoughts and emotions about those actions and outcomes, that we create meaning from these experiences" (p. 579). The ability to share coherent personal narratives is critical for building and maintaining friendships, physical and socio-emotional wellbeing, classroom participation, and success in academic and vocational settings. Moreover, personal narratives are important when describing and interpreting past experiences, for example, times when visiting the doctor or when describing a serious incident that happened at school.

Although the sharing of personal narratives is universal [5], little is known about how personal narratives are impacted by cultural differences. An understanding of the similarities and differences in the personal narratives of children from a diverse range of languages and cultures could increase their clinical utility for clinicians assessing language and communication across languages and cultures. Despite the importance of personal narrative proficiency for participation in society, research and clinical efforts to date have tended to focus on fictional rather than personal narratives [6, 7]. The current study addressed this gap in the research by developing a protocol for eliciting personal narratives in school-age children and evaluating its feasibility with 10-year-old children from 10 different countries, speaking 8 different languages.

### The development of personal narrative skills

Children develop personal narrative abilities during the preschool years, with parental reminiscing style playing an important role in fostering children's narrative development and autobiographical memory [8]. When sharing personal narratives, narrators tell the listener about an event that has happened to them and convey the meaning of that event to the listener [9]. A *coherent* personal narrative thus needs to include when, where, and what event took place, including the narrators' actions in a logical order, so that a naïve listener can make sense of the narrative. In addition, the personal narrative needs to convey what the event meant to the narrator [10, 11].

Researchers have charted a general developmental trend in personal narrative proficiency from pre-school into adolescence. For example, Peterson and McCabe [2] and McCabe [12]

found that 3-year-olds tended to produce two-event narratives; 4-year-olds produced more than two events but their events were often out of sequence; 5-year-olds were able to relate past events in a logical order and conveyed the meaning of the event (i.e., included an evaluation) without a resolution; by the age of 6, children produced what the researchers called a 'classic' narrative, containing at least two past events, a high point (evaluation), and a resolution. Reese et al. [11] investigated personal narrative coherence across a wider age-range, from pre-school into adulthood, focusing on three dimensions: context (orientation to time and place), chronology (the order of actions included), and theme (the meaning-making aspect of the narrative). Three-to-five-year-old children produced narratives that were on topic, but they often left out contextual information, and chronology was poor. School-age children provided some contextual information, and their performance on chronology improved. Chronology continued to improve from young to mid-adolescence, and by the time young people reached mid-adolescence they provided more specific contextual information. Finally, most of the young adolescents' personal narratives were on topic and elaborated, but they did not always include a resolution or link to other autobiographical experiences.

## Cultural variations in personal narrative development

Children tend to produce personal narratives that reflect not only the cultural style of their community (see [13] for a summary), but also its sociocultural norms [14]. Children generally start sharing personal experiences from 2 years of age, often in conversation with their parents. Parents scaffold these narrative interactions, providing the child with a basic overall structure. As predicted by Vygotsky's [15] sociocultural theory, this will become the prominent model used by the child when creating personal narratives, which means that children's personal narratives are likely influenced by their parents' narrative styles, values, and beliefs. Some parents use an elaborative or topic-extending style in which they embellish previously introduced topics, thereby lengthening the conversation. Others use a repetitive style in which they ask questions repeatedly, or a topic-switching style in which they introduce new topics frequently. Parents using either of these latter two styles may have shorter conversations about each event and provide less narrative structure. Some parents appear to invite their children's input more than others, and some appear to expect short factual reports as opposed to elaborate narratives (see [13]).

Cultural variations in how mothers support their child's narrative have also been reported. Choi (1992, cited in [16]) found that Korean mothers were unlikely to encourage their children to introduce their own topics or contribute information, whereas Canadian mothers were more likely to encourage narrative co-creation. Similar variations have been found between Japanese and European American mothers and their children, with Japanese mothers providing fewer evaluative comments in response to their children's narratives and requesting less detail than the European American mothers [17].

Children from some cultures (e.g., African American) may produce topic-associating as opposed to topic-centred personal narratives, in which children include several experiences into their personal narrative, as opposed to a detailed description of one experience [13]. Cultural styles also may affect event sequencing, inclusion of extensive background information (such as family connections) and the way in which the narrator evaluates the events [18, 19]. In addition, the choice of language may influence the personal narrative, particularly if children are attempting to produce a personal narrative in their second language. For example, in Spanish, the use of referencing is optional (with a tendency for using ellipses), which may be transferred into English and affect the perceived coherence of the personal narrative [20].

The purpose of the current study was not to compare narrative styles across cultures and/or languages; instead, the purpose was to develop a globally useful protocol for answering

research and clinical questions of many kinds. Therefore, one of our goals was to ensure the elicitation method would be standardized, yet flexible enough to provide opportunity for children from different cultures to produce personal narratives that would reflect their own cultural styles.

## Eliciting personal narratives

Personal narratives have been elicited by previous researchers in a variety of ways, with the overall objective to encourage children to share a meaningful experience [2]. For example, Peterson and McCabe [2] used a conversational map procedure in which children were provided with a short prompting narrative, before being asked "Did anything like that ever happen to you?" Children were encouraged to share one of their personal experiences, with the examiner simply encouraging them by using neutral sub-prompts such as 'uhuh' or 'tell me more'. Examples of prompt topics included car accidents, holidays, and illnesses. Peterson and McCabe found that successful prompts were those that encouraged children to talk about 'stand-out' experiences, as opposed to experiences that they engage in regularly, which are more likely to elicit scripts (i.e., generalizations about recurring events). The most successful prompt topics for eliciting lengthy narratives included trips, car wrecks, hospitalizations, and pets. Westerveld and Gillon [21] adapted this task by adding a series of photos accompanied by short prompting narratives to encourage children to share their experiences.

As described in Reese et al. [11], other elicitation methods include encouraging children to recount a memory of a past event that had been selected by their mothers, asking children to describe (recent) satisfying and disappointing personal experiences, asking children to recount negative experiences (associated with their health condition), or asking children to recall events that changed their lives and were still really important [22, 23]. Personal narratives can also be elicited using social problem-solving prompts, for example by asking the child about a time when someone asked them to do something they knew was not permitted [24]. Alternatively, children can be asked to provide personal narratives in response to open-ended emotion cues (e.g., "tell me about a time that you were really scared / frustrated / happy") [4]. However, a study by Fivush et al. [25] provided some interesting insights into children's personal narrative coherence when asked to narrate positive vs negative experiences. The researchers found that 5- to 12-year-old children who had been raised in violent communities produced more coherent narratives, containing more information about their thoughts and feelings, when talking about negative experiences. In contrast, when asked to tell narratives about positive events, these children included more information about people and objects and produced more descriptive detail.

In summary, a range of tasks have been used to elicit personal narratives from children of different ages, with no clear evidence that one task is more successful for monolingual English-speaking children. However, additional factors need to be considered when developing a protocol that can be used across cultures and/or with children who speak a language other than English.

## Challenges in developing a global protocol

When developing a personal narrative protocol for use across cultures and languages, the most important consideration is to avoid cultural and linguistic bias. We were concerned that adopting the conversational map procedure [2], which begins with the examiner providing a brief description of an event (e.g., hospital visit) as a model 'story,' could overly influence the child's response. Furthermore, it may be difficult to select events that are applicable across cultures, as not all children may identify with car accidents or ant-bites. Although photos have

been used successfully in Australia / New Zealand [26], the scenes (e.g., beach photo, theme park, holidays), would not be appropriate for use with children who live away from coastal waters or never go on school trips or holidays. More appropriate prompts may include open-ended emotion cues [4] or social problem-solving prompts [24] for eliciting narratives about meaningful events that may be experienced across cultures and regions. For the current project we have therefore opted for developing a set of six open-ended prompts tapping into different emotions, linking to both positive and negative experiences, characterized as (1) excited/happy; (2) worried/confused; (3) annoyed/angry; (4) proud; (5) problem situation; and (6) something important.

## Evaluating the clinical utility of a global protocol

Several questions guided this preliminary investigation into the feasibility of this global protocol, from now on referred to as the Global TALES (Talking About Lived Experiences in Stories) protocol. A successful global protocol should elicit personal narratives across cultures and languages. However, to allow for cross-country, cross-linguistic, and cross-cultural comparisons, creating a 'standard' protocol was paramount. Previous research has demonstrated how different elicitation conditions may influence children's narrative performance in a fictional context, such as the inclusion/absence of pictures or the use of a model story (e.g., [27, 28]). We therefore collaborated as an international team of speech-language pathologists, with expertise and experience researching child language, to discuss existing literature and our prior experiences within our own countries and cultures to develop the six prompts used in this study. We then gathered preliminary data to examine whether the protocol prompts elicited adequate verbal responses from the children in our various countries, which varied in language and cultural heritage. Evaluation of adequacy required that we come to consensus on key evaluation criteria and assumptions. This process involved several steps.

First, based on existing research, we hypothesized that measures of narrative productivity (e.g., number of utterances, number of words) should be fairly consistent when geographic location is the main variable [28], although we tempered this hypothesis based on differences found in adolescents' productivity (US vs Australia) in a persuasive discourse context [29]. Further, researchers in East Asian cultures have noted that extensive talking about oneself is discouraged, which suggests that cultural differences could result in shorter personal narratives for some [17].

Second, we expected that some protocol prompts would be more successful than others in eliciting responses and agreed that a way to evaluate this would be to ask about the number of follow-up prompts needed to encourage the children to produce a past-event narrative. Because previous cross-cultural research had indicated differences in how children from diverse cultures structure their personal narratives (e.g., topic associating vs topic-centered) (see [13]) in response to similar prompts, it was not clear whether all protocol prompts would be equally successful in eliciting a response from children with diverse cultural and linguistic backgrounds. It was also unknown if children in some cultures might require more prompting to share a past personal event with an adult (see [30], for an overview). Thus, some of our questions were exploratory.

Third, we considered that there might be variations in the topic of children's responses based on cultural values and beliefs that are associated with child-rearing goals and practices in different countries. To illustrate, studies have found variations in topics discussed by mothers and their kindergarten-age children between European American dyads and Hispanic dyads, with the European American dyads more likely to discuss child-peer comparisons [31]. Other research suggests that East-Asian dyads are more likely to talk about behavioral

expectations and social norms, compared to a tendency to focus on thoughts and feelings in European-American dyads [32, 33]. Taken together, a successful global protocol should elicit spoken language samples from children across languages and cultures, but it also should provide flexibility for children to choose their own topics in response to the prompts.

## The current study

This study was thus an initial investigation into the feasibility of a standard global protocol for eliciting personal narratives in school-age children across the world. Our main aim (in Part I of this two-part investigation) was to describe the variability in children's responses with respect to productivity (across the six protocol prompts and by protocol prompt), the amount of prompting needed, and the topics of children's responses. To reduce the number of variables, we recruited 10-year-old children from mid-socio-economic areas, who were performing well at school (i.e., who did not have a history of language-learning difficulties). Most 10-year-olds are in their fourth or fifth year of schooling, which is typically characterized as a transition point from 'learning to read' to 'reading to learn'. The participants, thus, match the age group described in the Progress in International Literacy Study (PIRLS; [34]), which investigates the reading comprehension skills of Year 4 students across 50 different countries every five years. Considering the importance of spoken language proficiency for reading success, we determined this was a suitable age group for this pilot project. Our second aim (Part II of this investigation) was to obtain feedback from the researchers involved in this study about the process of adapting the protocol for use in their country/language. The research questions were:

Part I

1. How do children perform on the Global TALES protocol across languages and cultures on measures of verbal productivity (number of utterances and number of words) in response to the six protocol prompts?

2. Are some protocol prompts more successful than others in eliciting responses, without the need for a scripted follow-up prompt (as per the protocol)?

3. What are the topics of children's responses across countries, languages, and cultures, in terms of their commonalities and distinctions?

Part II

4. What are the researchers' views on the process of adapting the Global TALES protocol for use in their own country/language?

## Part I: Performance on the Global TALES protocol

### Methods, Part I

**Participants.** Participants were recruited through the researchers' networks and through local school and community leaders. Researchers were members of the Child Language Committee of the International Association of Communication Sciences and Disorders (IALP) at the commencement of the study (although others were added later; see Acknowledgements). Inclusion criteria for the children were: a) aged between 9 years, 6 months and 10 years, 11 months, b) no history of speech and language difficulties, and c) currently not receiving specialist services (such as speech-language therapy). To control for possible socio-economic differences, we aimed to recruit children who attended schools considered to be located in a

middle-income area. To confirm children met these inclusion criteria, parents were asked to complete a brief demographic questionnaire (see https://osf.io/ztqg6/ for a complete copy of the project protocol). Table 1 provides an overview of the participant details, including the child's country, chronological age, relative income area of the child's school (low, middle, high), parents' highest levels of education, and family's income relative to the average income in their country. As shown in Table 1, there were 249 participants from 10 different countries, speaking 8 different languages. For transparency Cypriot Greek and Greek were considered the same language.

**Procedure.**  Ethics permission was obtained through the Human Research Ethics Committee at Griffith University (HREC; No: 2018/273) in Australia, with all other countries obtaining ethics permission from their respective universities (using a prior consent process). All parents provided written consent for their children to participate in the study; the children provided verbal assent at the start of the session.

Data collection commenced in 2019 before start of the global COVID 19 pandemic and finished in 2021. All children were seen individually (face to face) in a quiet location, either at the child's school, the university speech and language clinic, in the child's home, or at a community venue subject to parental preference. All examiners were qualified speech-language pathologists or speech-language pathology students under supervision, except for one linguist. Prior to the first session, all examiners viewed a demonstration video, read the administration manual, and practised administering the elicitation protocol with at least one child whose data were not included in the study.

*Task*. To elicit the samples, the examiners administered the Global TALES protocol (see https://osf.io/ztqg6 version 1). In this protocol, children first received an explanation about the task. Children were then asked to "tell a story" in response to each of the 6 topic prompts. If the child responded 'yes' to a protocol prompt, such as, "Can you think of a time when you felt excited or really happy?" the examiner then asked the child "Tell me a story about that!" If the child did not respond to the protocol prompt, the examiner used a scripted follow-up prompt. If the child only provided one or two sentences, the following generic encouragements were allowed: "Can you tell me more? Can you explain what you mean by that? Is there anything else you can tell me?" Finally, to encourage the child to continue talking, the examiner could use additional neutral encouragements, such as 'uhuh'. All protocol prompts were asked (read aloud by the examiner) in a set order and presented simultaneously in print on laminated cards or on a computer tablet.

*Translation*. The protocol was translated, when required, for use in non-English speaking countries by one of the researchers who was a native speaker of that language. When translating the protocol, the researcher sought to ensure that the child would be given the same instructions, protocol prompts, and scripted follow-up prompts as the original protocol; that the protocol prompts would be administered in the same order; and that the protocol prompts and scripted follow-up prompts would tap into the same key emotions or type of events and be culturally and linguistically appropriate. Most of the translated protocols can be downloaded from https://osf.io/ztqg6/.

**Transcription and analysis.**  All sessions were audio recorded for transcription and analysis purposes. All samples were transcribed in the native language by individuals experienced in transcribing language samples (including researchers, graduate students, a linguist, and undergraduate speech-language pathology students), using standard Systematic Analysis of Language Transcripts (SALT; [35]) conventions. All reformulations, repetitions, false starts, and filler words (e.g., *uhm*) were put in brackets and not included as part of the analysis. Utterance segmentation was based on communication units (C-units), defined as an independent clause with its modifiers. Only complete and intelligible utterances were counted as C-units;

**Table 1. Demographic data.**

| | AU | BR | Croatia | CY | GR | IL_A | IL_H | NZ | RU | TW | USA |
|---|---|---|---|---|---|---|---|---|---|---|---|
| N | 40 | 21 | 27 | 19 | 20 | 20 | 20 | 20 | 20 | 20 | 22 |
| F/M | 21/19 | 11/10 | 15/12 | 11/8 | 10/10 | 10/10 | 10/10 | 10/10 | 10/10 | 10/10 | 11/11 |
| Language | English | Portuguese | Croatian | Cypriot Greek | Greek | Arabic | Hebrew | English | Russian | Chinese Mandarin | English |
| Age [years; months] | 10;3 (0;4) | 10;5 (0;4) | 10;1 (0;4) | 10;3 (0;7) | 10;4 (0;4) | 10;1 (0;8) | 10;5 (0;5) | 10;4 (0;4) | 10;4 (0;4) | 10;3 (0;3) | 10;3 (0;5) |
| (SD) Range | 9;11–10;11 | 10;0–10;9 | 9;6–10;5 | 9;8–11;9 | 10;0–10;11 | 8;9–11;0 | 10;0–10;11 | 10;0–10;10 | 9;8–10;11 | 10;0–10;11 | 9;9–10;11 |
| **Socio Economic Status** | | | | | | | | | | | |
| Low | 0 | 9 (40%) | 1 (4%) | | 4 (20%) | | 2 (10%) | 2 (10%) | 6 (30%) | 0 | 2 (9%) |
| Middle | 28 (70%) | 10 (50%) | 24 (89%) | | 14 (70%) | 16 (80%) | 10 (50%) | 13 (65%) | 10 (50%) | 18 (90%) | 11 (50%) |
| High | 12 (30%) | 1 (5%) | 2 (7%) | | 2 (10%) | 4 (20%) | 8 (40%) | 5 (25%) | 4 (20%) | 2 (10%) | 5 (23%) |
| NR[a] | 0 | 1 (5%) | 0 | 19 (100%) | 0 | | 0 | 0 | 0 | 0 | 4 (18%) |
| **Parent education** | | | | | | | | | | | |
| Primary School | 0 | 7 (33%) | 1 (4%) | 0 | 1 (5%) | | 0 | 0 | 0 | 0 | 0 |
| High School | 6 (15%) | 3 (14.2%) | 10 (37%) | 6 (31.6%) | 6 (30%) | 3 (15%) | 2 (10%) | 9 (45%) | 6 (30%) | 4 (20%) | 4 (18%) |
| Trade qual | 2 (5%) | 0 | 3 (11%) | 2 (10.5%) | 4 (30%) | | 6 (30%) | 0 | 0 | 2 (10%) | 2 (9%) |
| Bachelor | 18 (45%) | 10 (47.6%) | 2 (7%) | 5 (26.3%) | 9 (45%) | 12 (60%) | 7 (35%) | 9 (45%) | 14 (70%) | 8 (40%) | 6 (27%) |
| Post-graduate | 14 (35%) | 0 | 11 (41%) | 6 (31.6%) | 0 | 5 (25%) | 5 (25%) | 2 (10%) | 0 | 6 (30%) | 6 (27%) |
| NR | 0 | 1 (4.8%) | 0 | 0 | 0 | 0 | 0 | 0 | 0 | 0 | 4 (18%) |
| **Relative income (based on parent responses)** | | | | | | | | | | | |
| Very low/low | 0 | 0 | 0 | | 6 (30%) | | 3 (15%) | 2 (10%) | 0 | 0 | 3 (13.5%) |
| Middle | 13 (32.5%) | 20 (100%) | 18 (67%) | | 12 (60%) | 20 (100%) | 10 (50%) | 13 (65%) | 20 (100%) | 18 (90%) | 8 (36%) |
| High | 23 (57.5%) | 0 | 8 (30%) | | 2 (10%) | | 5 (25%) | 5 (10%) | 0 | 2 (10%) | 6 (27%) |
| Very high | 2 (5%) | 0 | 0 (4%) | | 0 | | 2 (10%) | 0 | 0 | 0 | 1 (4.5%) |
| NR | 2 (5%) | 0 | 1 | 19 (100%) | 0 | 0 | 0 | 0 | 0 | 0 | 4 (18%) |

AU = Australia; BR = Brazil; CY = Cyprus; GR = Greece; IL_A = Israel Arabic speaking; IL_H = Israel Hebrew speaking; NZ = New Zealand; RU = Russia;

TW = Taiwan; USA = United States of America;

[a] NR = No response. SD = Standard Deviation

unfinished and interrupted utterances were not included. As per standard SALT conventions, elliptical responses were counted as separate C-units. The following productivity measures were calculated, either automatically using SALT (in AU, NZ, US) or manually:

- Total number of C-units (utterances), in response to each protocol prompt and for all protocol prompts combined.

- Total number of words, in response to each protocol prompt and for all protocol prompts combined.

**Transcription reliability.** Reliability of transcription was checked in a variety of ways. For transcription accuracy, including utterance segmentation, two methods were used. For most countries at least 20% of the samples were checked by a second researcher, with percent agreement > 85% for both utterance segmentation and transcription accuracy. In other countries, all transcripts were checked by the researcher and any disagreements were resolved prior to using the transcripts for analysis. The S1 Appendix provides an overview of the reliability process and the results by country.

**Topic coding and reliability.** The goal of qualitative topic coding was to identify the main topic of each personal narrative without forcing topics into a pre-determined set of codes,

which could have suppressed country-specific variations. To accomplish this, the following process was used:

1. Each country-based researcher was asked to independently assign a topic to each of the stories they collected (in response to the 6 protocol prompts) to answer the simple question, "What is this about?"

2. Each researcher also sent the written transcripts (uncoded) to the independent research assistant, a native English speaker from New Zealand (NT). These transcripts were uploaded to Google Translate for translation into English (when applicable). The research assistant then independently assigned a topic to each story (Phase 1) and recorded the following additional information: a) checked if the examiner used the scripted follow-up prompts; b) checked if the examiner used any additional prompting; c) noted if the child did not provide enough detail in their story for it to be coded (e.g., said "don't know," or produced a response limited to 1 or 2 C-units).

3. The research assistant compared the topics she assigned (Phase 1) to the topics assigned by the original researcher.

4. The research assistant met (online via Zoom) with each researcher to discuss any discrepancies in topic assignment. All disagreements were resolved. The final list of extended topics generated from this process, with examples, is attached in S2 Appendix.

5. To facilitate further investigation, a second level of categorical grouping was used in which this large number of codes was sorted into a more manageable number of categories [36]. Four authors (RL, NN, MW, CW), all of whom are fluent English speakers but from three different countries, created these collapsed categories by reviewing and discussing the 105 topic codes in S2 Appendix (ranging from 12 to 22 for the six protocol prompts) to arrive at the smaller set of 31 topic codes (ranging from 3 to 7 topic codes per protocol prompt), which were at a similar superordinate level of granularity. This was accomplished by grouping subtopics with synonymous meanings or exemplars of a larger category into the same superordinate category (see results section).

6. One member of the research team (NN) recoded the original topic descriptors into the reduced number of mutually exclusive categories using the category descriptors and exemplars.

7. A second member from a different country (RL) independently recoded 20 percent of randomly selected personal narratives across the 10 countries for reliability analysis. Simple percentage of agreement was 94.5% (342 agreements, 20 disagreements, 362 total topic codes). Discrepancies that occurred most often involved personal achievement vs personal growth (protocol prompts 2, 4, 6).

## Results, Part I

**Children's performance on the Global TALES protocol.** To answer the first question, we analyzed children's performance on the protocol on measures of verbal productivity and syntactic complexity across all 6 protocol prompts. As our focus was on evaluating children's performance on the protocol across languages and cultures, we did not perform any statistical analyses to compare countries. The results are listed in Table 2 and graphically displayed as boxplots in Figs 1–3. The mean number of utterances produced in response to the Global TALES protocol ranged between 43.7 (IL_A; Israel_Arabic speaking) and 80.79 (CY; Cyprus), but 9 out of 10 group means fell within a range of 18 utterances (45.8–63.8). Means for

**Table 2. Performance by country on measures of productivity.**

|  | AU | BR | Croatia | CY | GR | IL_A | IL_H | NZ | RU | TW | USA |
|---|---|---|---|---|---|---|---|---|---|---|---|
| N | 40 | 21 | 27 | 19 | 20 | 20 | 20 | 20 | 20 | 20 | 22 |
| **Utterances** |  |  |  |  |  |  |  |  |  |  |  |
| Mean | 68.75 | 51.10 | 45.81 | 80.79 | 63.80 | 43.70 | 50.40 | 55.75 | 62.30 | 61.00 | 54.09 |
| Median | 63.50 | 43 | 38 | 74 | 57.50 | 41 | 49 | 51 | 63 | 47.50 | 43.50 |
| SD | 27.49 | 37.63 | 27.31 | 47.06 | 22.91 | 10.61 | 12.86 | 20.57 | 32.78 | 45.59 | 36.67 |
| Min | 25 | 22 | 25 | 23 | 31 | 23 | 30 | 35 | 17 | 22 | 19 |
| Max | 123 | 199 | 169 | 201 | 126 | 71 | 80 | 114 | 131 | 227 | 185 |
| **Total Number of Words** |  |  |  |  |  |  |  |  |  |  |  |
| Mean | 622.45 | 427.29 | 353.48 | 396.68 | 428.35 | 201.90 | 385.45 | 182.00 | 475.60 | 518.10 | 160.09 |
| Median | 556.50 | 362 | 324 | 328 | 397.50 | 189 | 388 | 430 | 450 | 378.50 | 379.50 |
| SD | 267.84 | 384.90 | 194.42 | 251.71 | 171.35 | 56.66 | 150.52 | 43.20 | 261.55 | 481.55 | 68.45 |
| Min | 223 | 166 | 166 | 106 | 185 | 108 | 174 | 124 | 118 | 158 | 71 |
| Max | 1251 | 1918 | 1217 | 1123 | 842 | 316 | 668 | 277 | 1137 | 2348 | 365 |

AU = Australia; BR = Brazil; CY = Cyprus; GR = Greece; IL_A = Israel Arabic speaking; IL_H = Israel Hebrew speaking; NZ = New Zealand; RU = Russia;

TW = Taiwan; USA = United States of America

numbers of utterances were influenced by individual variation, which was high, ranging from one child who produced 17 utterances (RU; Russia) to another who produced 199 (BR; Brazil). As shown in Fig 1, only 12 outliers were observed across 6 countries, with Croatia showing the highest number of outliers (5). Reflecting the smaller grain size of words compared to utterances, even greater variability was observed in the total number of words, with the mean ranging between 160 words (USA) and 622 words (AU; Australia); individual variability ranged between 71 words (USA) and 1918 (BR). There were 10 outliers across 7 countries, with Brazil showing the highest number of outliers (3).

**Children's performance by protocol prompt.** Next, we investigated if some protocol prompts were more successful than others in eliciting responses, as measured by the number of utterances/words per story. As shown in the S1 Table, on average, all protocol prompts

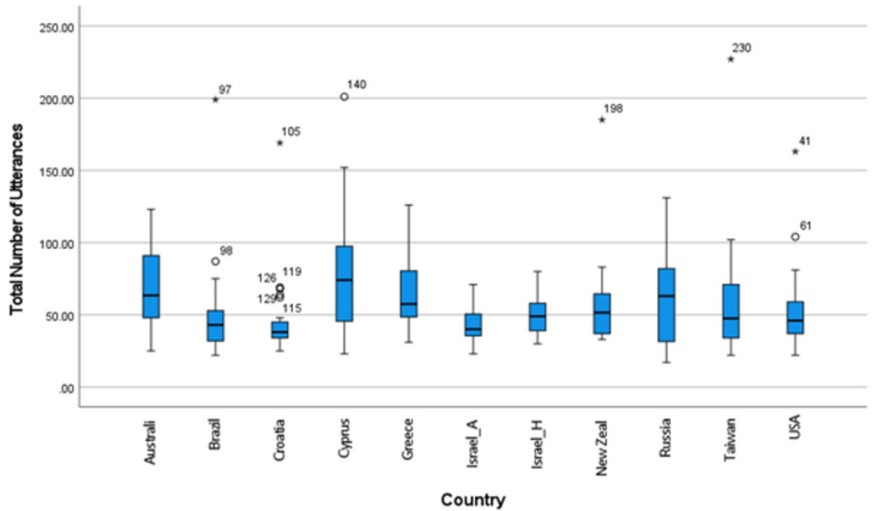

**Fig 1. Total number of utterances by country.** Indicates an outlier (more than 1 standard deviation [SD] from the mean; * outlier >2 SD from the mean; numbers refer to individual case numbers in SPSS.

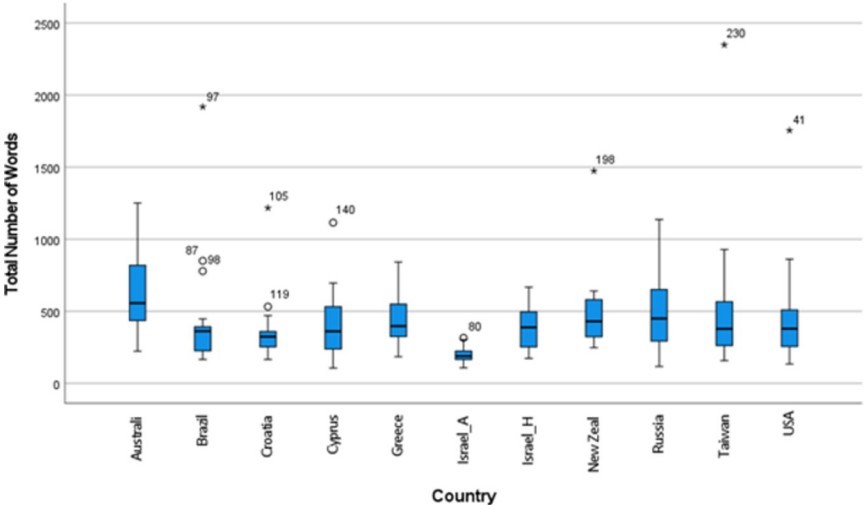

**Fig 2. Total number of words by country.** Indicates an outlier (more than 1 standard deviation [SD] from the mean; * outlier >2 SD from the mean; numbers refer to case numbers in SPSS.

elicited approximately 5 to 10 utterances. However, closer inspection revealed minimum scores of 0 utterances across all protocol prompts across five countries (AU, CY, IL_H, NZ, USA), indicating some children did not produce a response to some protocol prompts. Further analysis showed that out of the 1488 possible responses to protocol prompts (i.e., 248 participants x 6 protocol prompts), there were only 13 non-responses (0.87%), with protocol prompt 6 (something important to you) yielding 6 non-responses. Next, we inspected if scripted follow-up prompts (as per the task description) were needed to elicit a response. When considering all protocol prompts that elicited a response, across all countries / languages, the percentage of children who received a scripted follow-up prompt ranged between 34.3%

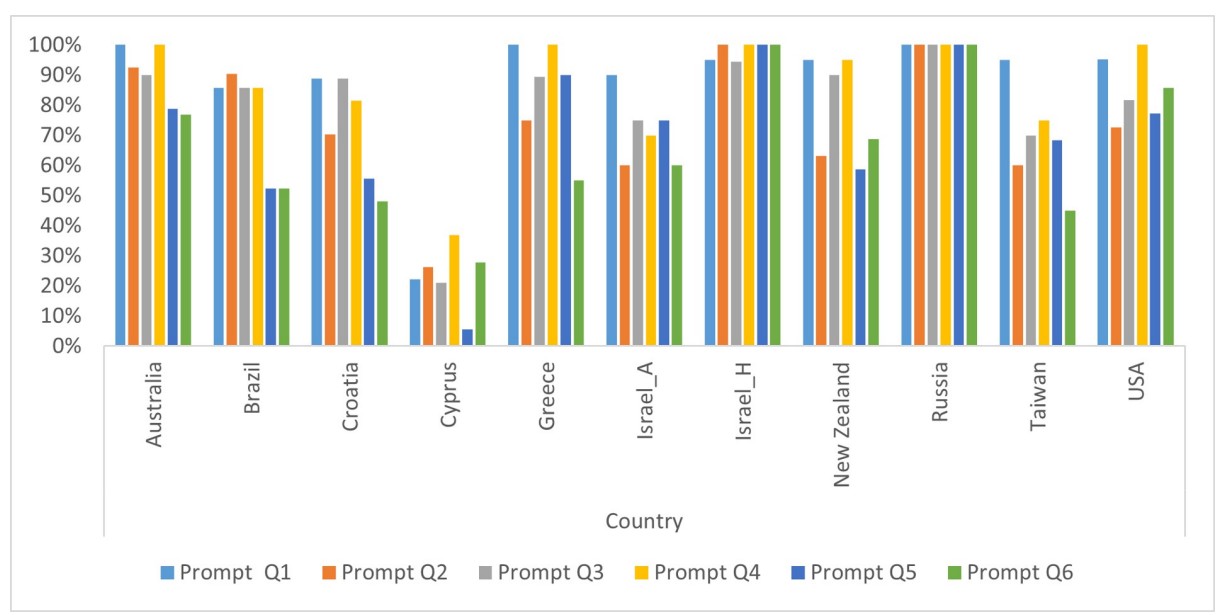

**Fig 3. Percentage of responses elicited without a scripted follow-up prompt, by country/language.**

(protocol prompt 6; important) and 10.6% (protocol prompt 1; excited or happy). However, when individual countries were considered, it was found that the most scripted follow-up prompts were used in Cyprus across the six protocol prompts (see Table 4). In contrast, no scripted follow-up prompts were used in Russia.

Finally, we inspected if the responses were codable for topic (see next section). Out of a possible total of 1488 responses, there were 79 non-codable responses (including 17 due to examiner error in presenting the intended prompt, see Table 5), amounting to 5.3% (4.1% when excluding those 17 responses). There were 18 responses to protocol prompts (possible total 160 per country) across two countries (BR and RU) that were non-codable, mainly because additional non-scripted prompts were given by the examiner. The total number of non-codable responses by protocol prompt ranged between 6 (protocol prompt 1) and 15 (protocol prompts 5 and 6).

**Topics of children's narratives.** To answer the third research question, we investigated the topics of children's responses across countries and cultures in terms of their commonalities and distinctions. Table 5 shows frequencies for the final topic codes that were collapsed from the original set of 105 topics codes to a more manageable set of 31 topic codes (ranging from 3, for protocol prompt 3, to a maximum of 7, for protocol prompt 6). Visual inspection of the top two frequencies for each topic by country / language group (with 2 language groups in Israel, for Arabic and Hebrew speaking) shows that most protocol prompts elicited similar topics across countries. For the first protocol prompt (excited), the most frequent topic for 10 of the 11 groups was a family event, such as a trip or holiday with the family. The second most frequent topic for protocol prompt 1 (excited) was about a new experience or item. For the second protocol prompt (worried), topics were more divergent. Most children in most countries chose a topic about school for their 'worried' stories, but there were exceptions. Russian and American children also told stories frequently about being worried about new challenges, such as moving or new skill challenges. Children in Brazil and Cyprus told stories most frequently about worries regarding illness, injury, or death, including death of a pet. Arabic-speaking children in Israel told stories most frequently about worries about losing someone or something. Hebrew-speaking children and Taiwanese children told stories most frequently about worries regarding families and friendships that were disrupted. Children in all 11 groups had the same most-frequent topic in their 'annoyed' stories, siblings and peers, and in their 'proud' stories, a personal achievement involving academics, athletics, or music. Children in 8 countries (all but Russia and Taiwan) told 'problem' stories (protocol prompt 5) most frequently about peer and family relationships that required fixing. In Russia, the problem topic was more frequently about achieving personal growth through apologizing, taking responsibility, or changing oneself. In Taiwan, problem stories most frequently talked about a problem at school involving forgotten homework or an upcoming test. In the USA, children told problem stories about needing to find or fix something equally frequently as about peer and family relationships. In response to the sixth prompt, about 'something important', the most frequent topics were personal achievement or a family event or support. The only exception was that the children in Taiwan told 'important' stories most frequently about personal growth by overcoming fear or being involved and helping others.

# Part II: Adapting the protocol for use across countries/languages: Researchers' views

## Methods, Part II

**Participants.** The research question for Part II of this study asked about the researchers' views on the process of adapting the Global TALES protocol for use in their own countries and

languages. This project was approved by Griffith University's Human Research Ethics Committee (HREC; No: 2018/273). Purposeful sampling was used to invite researchers who had participated in the Global TALES project and met either of the following two inclusion criteria: researchers who had plans to collect and analyze personal narratives from a sample of 10-year-old children in their respective countries, or researchers who had already collected and analyzed these personal narratives. Interviewees from 12 countries agreed to participate i.e., Australia, Belgium, Brazil, Croatia, Cyprus, Greece, Iceland, Israel, New Zealand, Sweden, Taiwan, and USA. Eight countries had one participant each, who were the lead investigators in their respective countries. For four countries, the interviewees included the lead investigator, as well as research assistants or students who had assisted with the recruitment, data collection, and/ or analysis. These included three participants each from Greece and Israel, and two each from Cyprus and Brazil. Country names were used to identify participants rather than assigning pseudonyms to individuals (with permission).

**Data collection.**   Data were collected using semi-structured interviews, and the team followed the COnsolidated criteria for REporting Qualitative studies (COREQ) [37] (see S3 Appendix for the completed COREQ form). Three members of the research team, MW (project leader from Australia), RL (research team member from Ireland with expertise in qualitative research) and NT (a qualified speech and language therapist and graduate researcher from New Zealand) designed the interview guide. Regarding reflexivity, the three members of this team were speech and language therapists who held positive views about the Global TALES project. To accommodate for this bias, they were careful to include open-ended questions and probes where participants could discuss both positive and negative experiences. The interview guide focused on four areas: experiences of translating the protocol and views on its cultural appropriateness in the participating countries; experiences of using the protocol to elicit narratives; experiences of the transcription and analytical processes; and views on next steps for the Global TALES project (see S4 Appendix for a copy of the interview guide).

Given that English was not the first language of many participants, key questions were sent to participants prior to the interviews. NT collected the data using a semi-structured interview format using the online platform Zoom. "In semi-structured interviews the researcher will have an interview guide but will be [more] flexible in encouraging the participant to talk openly and will explore or probe issues that the participants raise" [38]. Interviews were conducted in English, and group interviews were held where there was more than one interviewee in a country. The interviews were digitally recorded using the Zoom app and audio recorded using the iPhone 11 voice recorder/memos app. The audio recordings were transcribed using otter.ai; an online transcription program. Otter.ai automatically transcribed the files, which were then manually checked by NT for accuracy or errors and updated where needed.

**Data analysis.**   A member of the research team (RL) who is experienced in use of content analysis, first used content analysis to analyze the transcripts. Content analysis is defined as "a systematic and objective means of describing and quantifying phenomena" [39]. The outcome of the analysis is a set of categories that describe the topic of interest. In the first instance, a deductive approach was used where the interview topics were used as categories (in parent nodes) in NVivo12 [40] (i.e., translation, generating narratives, transcription, next steps). During the coding process, two additional categories were added inductively (i.e., involvement in the project and benefits/opportunities). The data were coded to subcategories inductively within these six categories in child nodes in NVivo12. These subcategories and categories were then reviewed, merged, and subsequently refined into four categories, each with subcategories. The participants were invited to review the analysis to check whether the results reflected their experiences. This member-checking was particularly important given that English was not the first language of many participants, and it was necessary to ensure that the analysis captured

their intended meanings. Three participants suggested some minor editorial amendments, and one participant provided a content clarification on the challenges of transcription.

## Results, Part II

The analysis revealed four categories, each with subcategories.

**1. Overall experiences of involvement in the Global TALES project.** *Positive experiences.* All participants (researchers) in all countries reported positive experiences of their involvement in the Global TALES project. They enjoyed and valued being part of an international project. Participants from six countries reported that they followed the protocol carefully to ensure consistency. Participants from eight countries valued the practical support they received from the Project Leader (MW) e.g., videos, response to queries etc.

*"I believe that . . . all the instructions about the sample was very clear, very helpful from the protocol." (Greece)*

*"But we did have the video [project leader] prepared that that helped a lot." (Cyprus)*

*"We had a very good support. [project leader] did it, very nicely. And we could ask her many questions." (Israel)*

*Value of personal narratives.* Participants from eight countries reported that they were pleased to see a focus on personal narratives, given the universal importance of narratives and the lack of data on personal narratives in all countries. Because people tell stories every day, some viewed the protocol as an ecologically valid tool and commented that children may feel less like they were being 'tested' when asked to tell stories (Belgium and New Zealand).

*"I think you can use it [the protocol] . . . you don't put any pressure on children. It's a normal story, normal talk. So even if children are not that communicative . . . they don't get the impression that that you are investigate talking to them" (Belgium)*

*"before I started . . . studying the personal narratives, I never, I had never stopped to think about the . . . the importance of this kind of narrative so I guess it will be. I think it will be very important here in Brazil". (Brazil)*

*"we don't have any protocol for . . . personal stories . . . I'm not quite sure that anybody ever asks children to tell a personal story. Up to now, so I would say this is the first time that we are actually involved in this personal story in how to encourage child to uh to produce such story . . . so it is excellent that we have a protocol in Croatia for this . . . type of narrative" (Croatia).*

*Challenges.* Some participants experienced challenges. Participants from two countries reported some challenges recruiting participants. For example, in Brazil the researchers were not based in schools and there were challenges with recruitment of participants.

*"I think one thing that is very important to keep in mind. That in Brazil SLPs [speech-language pathologists are] not in schools. So we don't have this straight access to children." (Brazil)*

Some reported difficulties with access to participants in schools where some schools were reluctant to engage in research activities (Australia). In the USA, concerns were expressed about the emotionally toned protocol prompts by one set of adoptive parents who were reluctant to provide consent for a child with a history of trauma that might be triggered by such

prompts. In some countries, the data collection was on hold because of the COVID-19 pandemic (Belgium and Sweden). Some countries experienced difficulty obtaining ethical approval and reported that it was helpful to have an ethics application that had been approved in another country (Brazil).

*Potential to develop new screening/assessment tool.* Participants from six countries talked about the lack of normative data on personal narratives in typically developing children in their respective countries. This project provided potential to use the data generated with typically developing children to inform the development of a new screening/assessment tool that could be used across countries. Some suggested that some of the protocol prompts could be used as a warm-up/rapport building activity (USA, Australia, and Cyprus). Participants from 10 countries reported that this project could serve as a foundation for developing a screening tool. Given that this was a feasibility study and participants collected data with 20 children, some talked about the need to collect data from larger, more representative samples, including children from different age groups as well as diverse linguistic and cultural backgrounds within their countries (USA, New Zealand). Three participants talked about the importance of ensuring that any norms generated were robust (e.g., reliability, validity, and sensitivity/specificity) (Greece, USA, and New Zealand).

> *"because of children's personal storytelling provides an important example of their spontaneous speech, it's a key tool for researchers . . . research . . . is very time consuming and requires checks to make sure that these tools are reliable . . . we can be sure that the tools are valid, that these, they measure what they are designed to measure and finally they can be used in other research on personal storytelling, eh providing more scientific data . . . the absence of such data in our in our country, and we think that this protocol should be very helpful." (Greece)*

> *"We need more tests in Hebrew. We don't have enough test with norms . . . I think that the protocol is not yet ready for use. We need much more data, for different ages. In order to be able to use it, and also, I think we have to test it on children with disabilities, such as DLD [developmental language disorder], ASD [autism spectrum disorder] . . . to try to find out its sensitivity". (Israel)*

*Further research opportunities.* Some were interested in exploring cross-cultural and within-culture similarities and differences. Participants from six countries expressed an interest in using the tool with children with dyslexia and/or language disorders. Participants from four countries expressed an interest in conducting more detailed semantic (e.g., lexical diversity) and syntactic analysis. One participant also suggested that student voice could be incorporated into the ongoing development of the tool, e.g., by asking children for their views about the protocol.

**2. Experiences of translation.** *Easy to translate.* All participants in the nine countries where translation of the protocol was required reported no difficulties with the translation process. Different approaches to translation were used. For example, the participants in Israel used a process of translating for meaning rather than using direct word-for-word translation. Others used the process of back-translation or checking the translation with others (Greece, Iceland, Brazil). Three respondents reported that the children had no difficulty understanding the instructions in the translated protocol.

*Addressing language-specific translation challenges.* Respondents from four countries reported language-specific translation issues where children may not be familiar with a word in the prompt in the protocol. For example, in Brazil, the word 'annoyed' was replaced by 'angry'. In Cyprus, the wording in the 'problem' story prompt was adapted to enable children

to understand it more easily i.e., "what created a problem for you and how you solved the problem". In Iceland, the children did not understand what the translation of the word 'worry' meant, and the interviewer re-phrased the protocol prompt to enable the children to understand. In Sweden, the translation of the word 'excited' required further explanation.

*"[name of researcher] and I and myself and one of our students we practiced [the translated version] on each other and we practiced on kids and piloted it and saw what we would get out of the children. So we ended up with a question . . . that made us all feel good. For example, the question that we are actually analyzing related to what made you, what created a problem for you and how you solved the problem". (Cyprus)*

*"For some of the children, the language was too complex, or, like the word 'worry' they didn't, the translation that I had used . . . because they didn't quite understand what I was talking about, which was 'worried'. So I reframed it, because they asked and said, 'What do you mean', and because it was interview, I could reframe it. So talk about . . . when something has happened that they thought was bad, or something had happened that they might have happened to them or something like that." (Iceland)*

*"So I was thinking a lot, how do I translate 'excited' . . . because it was not quite, some, straightforward translations had to use the Swedish word 'happy' . . . so I was thinking a lot about 'excited'. And what did you really mean by that? And I was thinking, maybe they will misunderstand me. So I used a few more, I explained it in more words, than excited. I think it's one of the questions." (Sweden)*

**3. Experiences of transcribing and analysis.** *Prior experience and quality control.* Participants from nine countries reported that they transcribed the transcripts manually and three reported that it was a time-consuming process. There was a range of experience among the participants. Although some participants reported that they had experience with different types of narrative analysis prior to their involvement in the Global TALES project, for many, transcription and analysis of personal narratives was new. Participants from nine countries valued the clarity provided in the instructions. Three participants reported that in instances where the data were collected by students, these data were checked by the lead investigators who were satisfied with the interviewers' skills. In some countries, students carried out the transcription and four described quality control processes in which the lead investigator reviewed the transcript to check for accuracy.

*"The student did the interviews and other students analyzed them . . . so she listened to that she watched and listened to the interviews and then has written it down. But she didn't use the suggestions when there was an utterance to use . . . another new line and she didn't use it or use it this kind of brackets for this thing. And so afterwards I have read the manual and seeing that it's very helpful." (Belgium)*

*Challenges in the transcription process.* Participants reported challenges with the transcription process in four countries. For example, in Croatia there were challenges with decisions regarding transcribing overlapping speech, revisions, which of the three different dialects to use, decisions about dependent and main clauses, and selection of C-units. In Cyprus, there were challenges with decisions regarding coding utterances vs sentences, managing revisions, incomplete sentences, and decisions about coding C-units and pauses. In Israel, there were challenges regarding the coding of C-units. In Greece, there were challenges regarding the number of different words because of linguistic differences. It was not possible to use SALT

because it does not support the Greek language. However, the team in Greece made adaptations to ensure that they could conduct comparable analyses in the Greek language samples. Participants in two countries would like an automated system to reduce the time required to transcribe and analyze the samples that could make the tool more attractive for clinicians.

> *"Sometimes it is very, tricky to transcribe some things, especially when we have overlapping in the speech, or when the children start to produce something and then interrupt, then somebody interrupt his sentences or, just stop to speak, or to pronounce some words, and then he cut the words." (Croatia)*

*Challenges analyzing the topics in the children's stories.* For many participants, analyzing the topics in the children's stories was new and interesting. However, participants in two countries reported challenges analyzing the topics in the stories and expressed concerns about consistency in this analysis across countries (Brazil and USA).

**4. Experiences generating data.** *Cultural acceptability.* All respondents reported that the translated protocol was culturally acceptable, albeit acknowledging that there was ethnic diversity in their respective countries that was not represented in their participant samples. Respondents in three countries reported that some of the questions in the demographic questionnaire for parents were not culturally appropriate e.g., questions about parental education levels and SES (Belgium, Croatia, and Greece). Some participants were uncertain about the cultural fit of the protocol for Aboriginal children. Some participants also commented that some of the scripted follow-up prompts might need to be tweaked to ensure a cultural fit. For example, one of the scripted follow-up prompts included talking about holidays which may not be appropriate for children who may not have had holiday experiences due to limited financial resources, or who do not celebrate certain religious holidays. Another example of a scripted follow-up prompt used in the worried-protocol prompt included a school project, which again, not all children may have experienced.

> *"The prompt for [the] happy [story] . . . I think this special holiday one is quite specific to [children from] . . . middle to upper socioeconomic backgrounds . . . I know of the children who were from . . . poor families, for example, they don't have a big vacation. That doesn't spark something in them hearing about this big fancy family vacation because they probably haven't gone on one . . . so probably a different 'excited' or 'happy' example that's not based around that might be better." (New Zealand)*

*Protocol prompt usefulness and challenges.* All participants reported that the six protocol prompts were useful in generating personal stories and no protocol prompts were deemed unsuitable. The researchers used the scripted follow-up prompts and generic / back-channel prompts provided in the protocol to encourage children to provide longer responses. However, all respondents reported that some children provided short responses to the protocol prompts (e.g., "I don't know," "I'm not worried about anything") and required encouragement and further prompting to generate longer samples. Some reported that the best protocol prompts to generate a narrative was the problem-prompt because the others tended to generate short descriptive answers rather than a 'story' (New Zealand). The respondents in Greece and Cyprus reported that the children tended to talk about the problem in the 'problem' narratives rather than the solution and required more specific prompts to address this part of the story. One participant also brought up the possibility that the problem-prompt may generate disclosures, so it is important that the interviewer is prepared to deal with this if it arises (Australia). Another participant talked about including this in the ethics application (Croatia).

*"The only problem that we had, actually was that our ethical board wanted that we develop . . . how to react, how to deal in some unexpected situations . . . Like, what to do. If some question actually be a trigger for some unexpected stories, like a violence in family, or, and what to do. And that was recommendation of our ethical board, and we needed to develop this protocol." (Croatia)*

Participants from six countries wondered whether or not cultural factors or the type of prompts may have influenced the length of the narratives generated. For example, the Brazilian participant reported that while the 'annoying' prompt appeared to be the easiest, the children needed time to respond to the 'proud' prompt. The New Zealand participant also reported that it was difficult to generate a narrative with both the 'proud' and 'something that was important to you' protocol prompts. She felt that this may be a cultural issue, in which some children would not typically talk about what made them proud. Likewise, participants in Iceland and Taiwan reported that it might not be typical for children to talk about their feelings.

*"For our children . . .that's true for some questions is not . . . very open I mean, it's not very common for our children to talk about, in their daily life . . . For example, like our children they don't talk to our strangers like the people not familiar with them, talk about something they are worried about. But I think though the happy story . . .they can talk about it." (Taiwan)*

*"But I think, I don't know whether it's a cultural thing or just a difficult question thing. But the one about 'something that's important to you didn't really get much of a response. I don't know whether that's because it was culturally difficult or just because it's a difficult one." (New Zealand)*

*"Out of all of the questions, the one about being worried about something, I thought was probably the one that they felt most uncomfortable, probably answering the other ones that had more of a positive psychology slant, you know, when were you proud of yourself? Tell me a time when you're excited, really easy, really easy questions to ask kids. Again, probably those ones that provide a bit more of . . . perhaps a deeper response, which is probably when you're going to get quite a good language sample as well . . .I don't think that removing the question is necessary." (Australia)*

Based on the literature on narratives, one participant expected longer narratives in response to the 'problem' prompts given that there would be more to discuss. However, the responses were shorter than expected and the participant wondered whether this was due to context (e.g., the lack of spontaneity) because the children were asked to tell specific stories in response to protocol prompts.

*"I think that asking 'tell me a story about a time' when [there is] is a specific time is then you get an, more likely to get an anecdote where the child is actually recalling a particular event . . .I knew the problem prompt would work best or I felt like our experience would show that it would be most likely to elicit a true story using the story grammar, the traditional kind of Western culture story grammar, because if there's a problem, there's something to respond to and otherwise you sometimes get a series, or you get labelling. But I felt like most of these prompts, they seemed fairly short to me, the kids' responses. And I felt like I should be trying to get more from them. And that, but then, that of course backfires . . . then you no longer are looking at what they spontaneously give you, so I think we have to continue to look at that." (USA)*

Participants from two countries wondered whether children needed more time and familiarization with the task before the data collection session or time at the beginning of the session to think about the stories they might tell in response to prompts. One participant wondered whether visual stimuli may help children to think of a broader range of topics (children mainly focused on school topics) (Australia).

*"I just think, actually preparing them, giving them some time. And so, I would say build rapport. And I would say give them time, with pre-preparation, if you're going to ask those questions that might be a bit more challenging. Yeah, I think making the parents aware that you're going to ask those questions is probably an appropriate thing to do." (Australia)*

## Discussion

This study investigated the feasibility of the Global TALES protocol developed to elicit personal narratives from school-age children across a range of countries, languages, and cultures. The protocol contained six prompts with scripted follow-up prompts that tapped into a range of emotions and events. In Part I of the study, a total of 249 children from 10 countries, speaking 8 languages participated in the project. We examined if the protocol was successful in eliciting extended spoken language samples, based on measures of productivity, and whether children were more likely to produce responses to some protocol prompts more than others, based on number of utterances produced per story and number of scripted follow-up prompts provided by the data gatherers. We also investigated the topics of children's personal narratives by coding topics into categories for each of the six prompts and identifying topics that were used most frequently per country / language group. In Part II of the study, qualitative analysis techniques were used to look for patterns in interview data that were obtained from researchers participating in the project, based on their experience in using the Global TALES protocol.

### Productivity in response to the six protocol prompts

We first analyzed children's verbal productivity (number of utterances and number of words) in response to the six protocol prompts (with responses to all prompts combined). At group-level and across countries, children produced between 43 and 80 utterances, indicating the protocol was successful in eliciting language samples that may potentially be of sufficient length for assessment purposes, especially if part of a comprehensive assessment of children's oral language skills, even though Heilmann et al. [41] suggest a sample length of >50 utterances may be required for more detailed linguistic analysis. Although not the aim of this study, variability in performance between countries / languages was tentatively expected based on previous research with adolescents [29]and investigations of verbal productivity in children from different cultural backgrounds [17]. At the individual level, however, there was wide variability in performance with 'outliers' (i.e., children whose performance was at least 1 SD above or below the mean performance compared to their peers in that country) observed across six countries. In countries with low-scoring outliers, median scores were below the mean, indicating the effect these outliers may have on overall performance data. Variability in performance on spontaneous language tasks is common in children and adolescents [26, 42], although not always reported [11], and it reflects the relatively small samples and the inherent variability of spontaneous language sampling [43]. The next step in evaluating the Global TALES protocol for clinical purposes is further inspection of individual countries' results with larger and more representative samples of children and adolescents to examine the potential for gathering sufficient data for local norming purposes.

## Productivity by protocol prompt

To help the research team refine the protocol, our second research question asked whether some protocol prompts were more successful than others in eliciting responses, as measured by the number of utterances/words. Our results clearly indicated that all protocol prompts were effective in eliciting responses from most participants, across countries. These results indicate that although some prompts were not successful in encouraging a few children to verbally share a personal experience (i.e., no response was provided), this occurred in less than one percent of the stories prompted. Overall, the protocol was successful in eliciting six stories from almost all children. We also investigated if the scripted follow-up prompts (as per the task description) had been used to elicit the personal narrative. The numbers of scripted follow-up prompts that were used were remarkably similar across protocol prompts, ranging from 10.6% (prompt 1, excited) to 34.3% (prompt 6, excited), with all countries combined (see Table 3). However, closer inspection of individual countries (see Table 4) revealed some interesting differences, with Russia and Israel (Hebrew speaking) providing no, or very few scripted follow-up prompts, whereas Cyprus provided more scripted follow-up prompts than many other countries. Further inspection of the transcripts (including child and examiner verbal behaviours) is needed to better understand if more extensive prompting is in line with specific cultural socialization practices in those countries [30], or whether these differences simply reflected individual differences on the part of the adults gathering the samples.

## The topics of children's narratives

To answer our third research question, we investigated the topics children talked about in response to the six protocol prompts. As supported by the results for topic frequency in Table 5, our primary conclusion is that children around the world seem to respond to many of the protocol prompts with similar topics, but we caution against overgeneralization. The strong global similarities in topics elicited by the prompts, as well as the examples in which children in one or more countries talked about topics that were unique to those countries both deserve further investigation. We also caution that our sample sizes are relatively small and may not generalize to the wider population, and that second-level coding by a person from a different culture could have obscured important but subtle differences that only persons from the original culture would identify. On the other hand, some differences in topic frequencies for individual prompts by country may reflect true cultural differences (i.e., in beliefs, values, norms and practices), along with differences in how parents across cultures influence their children's personal narratives. For example, Schick and Melzi [30], in their review of oral narrative development in young children from diverse sociocultural backgrounds, observed that mothers in East Asian cultures talked more about behavioral expectations and social norms with their children, compared to European/American mother-child dyads who tended to

**Table 3. Percentage of responses elicited with or without a scripted follow-up prompt, all countries combined.**

|  | n | % with scripted follow-up prompt given | |
|---|---|---|---|
|  |  | No | Yes |
| Protocol prompt **1 (excited)** | 245 | 89.4 | 10.6 |
| Protocol prompt **2 (worried)** | 242 | 74.8 | 25.2 |
| Protocol prompt **3 (annoyed)** | 244 | 81.6 | 18.4 |
| Protocol prompt **4 (proud)** | 228 | 86.0 | 14.0 |
| Protocol prompt **5 (problem)** | 238 | 69.7 | 30.3 |
| Protocol prompt **6 (important)** | 239 | 65.7 | 34.3 |

**Table 4. Percentage of responses elicited without a scripted follow-up prompt, for each protocol prompt, by country.**

| Protocol Prompt | Country | | | | | | | | | | |
|---|---|---|---|---|---|---|---|---|---|---|---|
| | AU | BR | Croatia | CY | GR | IL_A | IL_H | NZ | RU | TW | USA |
| 1 | 100.0% | 85.7% | 88.9% | 22.2% | 100.0% | 90.0% | 95.0% | 95.0% | 100.0% | 95.0% | 95.2% |
| 2 | 92.5% | 90.5% | 70.4% | 26.3% | 75.0% | 60.0% | 100.0% | 63.2% | 100.0% | 60.0% | 72.7% |
| 3 | 90.0% | 85.7% | 88.9% | 21.1% | 89.5% | 75.0% | 94.4% | 90.0% | 100.0% | 70.0% | 81.8% |
| 4 | 100.0% | 85.7% | 81.5% | 36.8% | 100.0% | 70.0% | 100.0% | 95.0% | 100.0% | 75.0% | 100.0% |
| 5 | 78.9% | 52.4% | 55.6% | 5.6% | 90.0% | 75.0% | 100.0% | 58.8% | 100.0% | 68.4% | 77.3% |
| 6 | 76.9% | 52.4% | 48.1% | 27.8% | 55.0% | 60.0% | 100.0% | 68.8% | 100.0% | 45.0% | 85.7% |

AU = Australia; BR = Brazil; CY = Cyprus; GR = Greece; IL_A = Israel Arabic speaking; IL_H = Israel Hebrew speaking; NZ = New Zealand; RU = Russia; TW = Taiwan; USA = United States of America

focus on thoughts and feelings. We also observed that the use of the scripted follow-up prompts could influence the child's narrative topic choice by providing a more specific topic (see S2 Appendix) and that the timing of sampling relative to emotionally toned events, such as a global pandemic, could influence topic choices. This phenomenon was observed when sampling had to be suspended due to the COVID pandemic, and data collection begun again too late for the data to be included in the current analyses (Ireland), but when gathered, included frequent topics associated with pandemic experiences. Taken together, we found that children from around the world share many commonalities regarding topics of conversation. However, individual variability was still high, with more than 100 different topics initially identified. When evaluating the protocol with respect to desired flexibility, the prompts used in the Global TALES protocol thus seem effective in prompting children to share their past personal experiences without forcing them to focus on one particular topic.

## Researcher feedback on the process of adapting the Global TALES protocol for use in their own country/language

In Part II of this study, all participants provided positive feedback on their involvement in the Global TALES project. They highlighted the value of resources to support them in following the protocol, including written instructions and video recordings. In addition, many (eight) reported on the importance of research focussing on personal narratives, given the ecological validity of personal narratives and the lack of normative data in many countries. However, the importance of continuing to collect more data, including from a broader range of participants was highlighted, and some of the challenges of conducting research with school-aged children were mentioned, particularly during the global COVID pandemic. Of specific relevance to this feasibility study, many participants identified that the Global TALES protocol could be incorporated into existing assessment batteries.

One of the challenges with this project was developing a protocol that has utility across a range of different languages and cultures. It was not surprising that participants shared challenges around translating the protocol into other languages. Sometimes this related to concepts that are not easily expressed in a language, and therefore lack a tightly synonymous vocabulary item, as occurred for the protocol prompt focussed on sharing an example of a time you were "excited" in Swedish, which needed to be translated with a word for 'happy' and then further explained. However, all participants felt that the final protocol was appropriate for use in their context.

**Table 5. Summary of refined topic codes by country for the six protocol prompts, marked to show topics with highest** ** and second highest* frequencies.**

| Protocol Prompt | Topic | Examples | Frequency by Country[1] | | | | | | | | | | |
|---|---|---|---|---|---|---|---|---|---|---|---|---|---|
| | | | AU | BR | Croatia | CY | GR | IL-A | IL-H | NZ | RU | TW | USA |
| | | | n = 40 | n = 21 | n = 27 | n = 19 | n = 20 | n = 20 | n = 20 | n = 20 | n = 20 | n = 20 | n = 22 |
| **1. Excited** (18 original codes collapsed into 5) | Family event | Family trip, holiday, visit to theme park, other family activity; sibling activity | 20** | 11** | 12** | 7** | 9** | 6* | 11** | 15** | 7** | 8** | 17** |
| | New experience or item | Moving/relocating; getting a new pet; receiving a gift; animals | 10* | 8* | 8* | 6* | 2 | 10** | 1 | 4* | 5* | 3 | 3* |
| | Personal achievement | Personal, academic, sporting, or musical achievement; learning new skill | 5 | 1 | 5 | 3 | 3 | 2 | 4* | 1 | 3 | 5* | 1 |
| | Personal growth/ contribution | Being brave helping others; personal goal; independence | 0 | 0 | 0 | 1 | 0 | 0 | 2 | 0 | 0 | 0 | 0 |
| | Peer relationship | Building new relationship; reunion; time with friend; friend's birthday; social event | 5 | 0 | 2 | 1 | 6* | 2 | 2 | 0 | 3 | 4 | 0 |
| | | No codable response | 0 | 1 | 0 | 1 | 0 | 0 | 0 | 0 | 3 | 0 | 1 |
| **2. Worried** (22 original codes collapsed into 6) | School task | Academic task, forgetting homework, upcoming test, unknown expectations | 12** | 3* | 11** | 0 | 7** | 6** | 5* | 6** | 6** | 7* | 6* |
| | New challenges | Moving or relocating; performance or skill level worries; uncertainty of demands | 10* | 0 | 6 | 0 | 5* | 0 | 0 | 5* | 6** | 3 | 10** |
| | Safety concerns | Family safety; personal safety; having something stolen | 4 | 3* | 0 | 1 | 0 | 1 | 0 | 0 | 0 | 0 | 0 |
| | Illness, injury, or death | Family illness; broken arm, hospital visit; sick grandparent; pet illness, pet died | 5 | 10** | 7* | 9** | 1 | 3 | 2 | 1 | 2 | 2 | 5 |
| | Family/ friends relationships | Fighting; responsibility for damage; disappointing someone, being excluded | 3 | 2 | 0 | 3 | 5* | 4 | 7** | 3 | 3 | 8** | 0 |
| | Losing someone or something | Misplacing a toy; losing mum shopping, searching for something or someone lost | 6 | 2 | 3 | 5* | 2 | 6** | 3 | 4 | 1 | 0 | 1 |
| | | No codable response | 0 | 1 | 0 | 1 | 0 | 0 | 3 | 1 | 2 | 0 | 0 |
| **3. Annoyed** (20 original codes collapsed into 5) | Sibling/peer relationships | Being bullied or seeing others bullied; being ignored, copied from, lied to, or stolen from; siblings or friends fighting | 24** | 15** | 13** | 17** | 14** | 13** | 15** | 13** | 8** | 15** | 19** |
| | Parental issues | Permission refusal or plans cancelled; being ignored; parents fighting; punishment | 2 | 0 | 3 | 1* | 5* | 0 | 0 | 0 | 3* | 4* | 1 |
| | Expectations of school/ others | School/academic expectations; not understanding; unprepared; skill issues | 1 | 0 | 6* | 0 | 0 | 2 | 1* | 3 | 3 | 0 | 2* |
| | Personal frustration | Unable to find or get something; damaged item or toy; device dead; day wrecked | 11* | 2* | 5 | 1 | 1 | 4* | 0 | 4* | 4 | 0 | 0 |
| | Injury/illness | Injury, illness, or medical condition | 2 | 0 | 0 | 0 | 0 | 1 | 0 | 0 | 0 | 1 | 0 |
| | | No codable response | 0 | 4 | 0 | 0 | 0 | 0 | 4 | 0 | 2 | 0 | 0 |
| **4. Proud** (12 original codes collapsed into 3) | Personal achievement | Academic, sporting, or musical achievement; new skill; | 33** | 14** | 25** | 18** | 15** | 15** | 15** | 14** | 10** | 19** | 4** |
| | Personal growth or contribution | Brave; overcoming fear; honesty; social achievement; helping someone; personal relationship; initiative; finding something | 7* | 3* | 1* | 1* | 5* | 5* | 5* | 5* | 4* | 1* | 1* |
| | Achievement involving others | Sibling being born; family achievement; injury/illness/medical | 0 | 0 | 1* | 0 | 0 | 0 | 0 | 1 | 2 | 0 | 0 |
| | | No codable response | 0 | 4 | 0 | 0 | 0 | 0 | 0 | 0 | 4 | 0 | 17# |

**Table 5.**  (Continued)

| Protocol Prompt | Topic | Examples | Frequency by Country[1] | | | | | | | | | | |
|---|---|---|---|---|---|---|---|---|---|---|---|---|---|
| | | | AU | BR | Croatia | CY | GR | IL-A | IL-H | NZ | RU | TW | USA |
| | | | n = 40 | n = 21 | n = 27 | n = 19 | n = 20 | n = 20 | n = 20 | n = 20 | n = 20 | n = 20 | n = 22 |
| **5. Problem** (16 original codes collapsed into 5) | Peer/family relationships | Resolving conflicts, differing opinions; being copied; standing up to bullies; family support | 17** | 5** | 11** | 9** | 8** | 13** | 11** | 11** | 2 | 2 | 9** |
| | Finding or fixing | Finding or fixing something; replacing something; damaging something | 8* | 3 | 1 | 4* | 0 | 3* | 2 | 2* | 4* | 1 | 9** |
| | Personal growth or contribution | Apologizing; fixing a mistake; taking responsibility; changing mental set or learning new skill; overcoming fear | 0 | 3 | 4 | 0 | 3 | 0 | 3* | 0 | 8** | 3* | 2 |
| | Safety or wellness | Overcoming illness or injury, sibling wellness, family illness/injury | 5 | 3 | 5* | 1 | 4 | 3* | 2 | 2* | 0 | 0 | 1 |
| | Problem at school | Forgetting/doing homework, upcoming test, not understanding, forgetting tools | 8* | 4* | 5* | 2 | 5* | 1 | 2 | 2* | 3 | 14** | 1 |
| | | No codable response | 2 | 3 | 1 | 3 | 0 | 0 | 0 | 3 | 3 | 0 | 0 |
| **6. Important** (17 original codes collapsed into 7) | Personal achievement | Academic or sporting achievement; working as team; musical achievement | 20** | 5* | 22** | 5** | 8** | 7** | 15** | 7** | 8** | 4* | 7* |
| | Family event or support | Family birthday or reunion; family support; expressions of love; family trip or holiday | 8* | 7** | 2* | 5** | 4* | 3 | 1 | 6** | 1 | 1 | 11** |
| | Cultural | Participating in cultural activities; religious event; keepsakes; movie idol | 3 | 0 | 0 | 0 | 1 | 3 | 0 | 1 | 0 | 1 | 0 |
| | Personal growth or contribution | Overcoming fear; helping others; being involved; pet concerns | 1 | 0 | 1 | 4* | 1 | 2 | 0 | 3 | 2* | 6** | 0 |
| | Peer relationships | Fixing relationships; making a new friend; feeling closer; social achievement | 3 | 0 | 0 | 0 | 1 | 0 | 4* | 0 | 2 | 3* | 2 |
| | Safety and wellness | Calling ambulance; finding safer way home; overcoming injury; medical achievement | 3 | 0 | 2* | 3 | 4* | 0 | 0 | 0 | 2 | 2 | 0 |
| | New items or experiences | Receiving new pet or gift; surprise trip | 0 | 4 | 0 | 1 | 1 | 5* | 0 | 0 | 1 | 3 | 2 |
| | | No codable response | 2 | 5 | 0 | 1 | 0 | 0 | 0 | 3 | 4 | 0 | 0 |

AU = Australia; BR = Brazil; CY = Cyprus; GR = Greece; IL_A = Israel Arabic speaking; IL_H = Israel Hebrew speaking; NZ = New Zealand; RU = Russia;

TW = Taiwan; USA = United States of America; [#] the wrong prompt was provided.

## Limitations

This study has yielded some important, but preliminary results about the feasibility of a global protocol for eliciting personal narratives. We acknowledge that there are several limitations. Most importantly our findings are based on a relatively small number of participants from each country, most of whom came from middle socio-economic backgrounds and were aged between 9 and 11 years. Although this was based on deliberate decisions to limit within-country variability, it means that it is unclear if these results would generalize to a wider population. Future work should aim to recruit larger, more representative samples from each country.

Investigating language performance across a range of languages is complex, and as a result we only included productivity measures (total number of utterances and total number of words). Future studies might consider additional language measures to investigate grammar and use of semantics. The study is limited in choice of countries/languages, which was opportunistic, as most of the researchers were past or present members of the child language

committee of the International Association of Communication Sciences and Disorders (IALP). This has resulted in a relative over-representation of English-speaking children (3 of the 10 countries), with only one country from Asia. However, since starting this project, researchers from other countries have joined our project and many of them have started their own data collection, including researchers in Iceland, Sweden, Ireland, Korea, Belgium, Poland, and South Africa. Despite these limitations (and because of them), the current report should be seen as a work in progress and an attempt to cast a wider global net for more countries to be involved in data gathering.

## Future directions and clinical implications

Despite our promising results, we acknowledge that using a standardized protocol with scripted follow-up prompts may not be readily achievable across all cultures. In an attempt to avoid 'leading' the child to create a personal story, the examiner takes a fairly passive role when administering the Global TALES protocol. However, in some cultures children may expect stories to be jointly constructed, for example, and further research is needed to evaluate ways of engaging children in story telling in a culturally responsive, yet standardized way to allow for cross-cultural comparisons. Future research should also investigate the effects of specific prompts and/or conversation partners on children's ability to provide a coherent personal narrative across different cultures, providing much-needed information to enhance cross-linguistic and cross-cultural understanding of language development and disorders. Findings of such research may indicate that, in some cultures, it may not be appropriate for children to discuss feelings and/or talk about certain topics with unfamiliar adults. Regardless, using a standard protocol to collect normative data across a range of languages and cultures could allow researchers to learn more about this universal discourse genre, its development with age or years of schooling, and how culture may impact performance. These findings will also be important for clinicians who work with children from culturally and/or linguistically diverse backgrounds on a daily basis. Eventually, country-specific data, reflecting differences within and across countries and cultures and evaluated for reliability and validity, could result in a practical tool for timely identification of developmental language disorder. From a clinical perspective, having access to an ecologically valid tool will provide important insights into the impact of a child's communication impairment on their functioning and participation at home, at school, and in the community (see [44]).

## Overall conclusions

In summary, the results from this initial investigation into the feasibility of the Global TALES protocol for eliciting personal narratives in 10-year-old children from around the world are promising. The six protocol prompts were effective in eliciting discourse samples from most children, with no clear indication that one protocol prompt was more effective than another in eliciting responses. Children talked about a range of topics with clear commonalities and some differences across countries/languages, indicating the protocol provides enough flexibility for children to talk about experiences that were meaningful to them. In response to researcher feedback, which was generally positive, some minor changes to the wording of the protocol have been made. (Version 2 can be found in S5 Appendix or can be downloaded from https://osf.io/ztqg6/). These changes mainly relate to minor wording changes and some clarification to further standardize the number and type of scripted follow-up prompts that are allowed to encourage children to start their personal narrative and/or to continue talking, while maintaining the sense of spontaneity.

We now invite clinical researchers from around the world to join in conducting further research into this important area of practice to obtain a better understanding of the development of personal narratives from children across different languages and cultures and to begin to establish local benchmarks of typical performance. In the meantime, we welcome clinicians and educators to start using the protocol and provide feedback into its usefulness by contacting one of the authors. We trust this collaborative open-science approach to data collection and data sharing will help move the field forward with the ultimate aim of promoting successful communication.

## Supporting information

**S1 Appendix. Transcription reliability by country.**
(DOCX)

**S2 Appendix. Topics Part I.**
(DOCX)

**S3 Appendix. Consolidated criteria for reporting qualitative studies (COREQ): 32-item checklist (Tong et al., 2007).**
(DOCX)

**S4 Appendix. Global TALES interview guide.**
(DOCX)

**S5 Appendix. Global TALES protocol v2.**
(DOCX)

**S1 Table. Productivity measures by country.**
(DOCX)

**S1 Fig.**
(JPG)

## Acknowledgments

We are deeply grateful to the children who participated in our study and their families, teachers or community leaders who supported their participation. We also thank our colleagues and research assistants that supported us in undertaking this research. We sincerely thank Nikki Turpie (NT), speech-language therapist from the Child Well-being Research Institute in New Zealand for her help in organizing the data and conducting the research interviews in Part II of this study. We also wish to acknowledge Dr. Cristina McKean, from the United Kingdom, for her assistance in developing the project and her work piloting the initial protocol.

**Additional members of the Global TALES consortium**

Jóhanna T. Einarsdóttir (University of Iceland, Iceland), Sharon Moonsamy (University of Witswatersrand, Johannesburg, South Africa), Ann Nordberg (University of Gothenburg, Sweden), Christel van Vreckem (Artevelde university of applied sciences, Belgium).

## Author Contributions

**Conceptualization:** Marleen F. Westerveld, Nickola Wolf Nelson, Sara Ferman, Gail T. Gillon, Carol Westby.

**Data curation:** Marleen F. Westerveld, Nickola Wolf Nelson, Sara Ferman, Jelena Kuvač Kraljević, Kakia Petinou, Eleni Theodorou, Tatiana Tumanova, Ioannis Vogandroukas, Carol Westby.

**Formal analysis:** Marleen F. Westerveld, Rena Lyons, Nickola Wolf Nelson, Carol Westby.

**Funding acquisition:** Marleen F. Westerveld, Gail T. Gillon.

**Investigation:** Marleen F. Westerveld, Rena Lyons, Nickola Wolf Nelson, Kai Mei Chen, Mary Claessen, Sara Ferman, Fernanda Dreux M. Fernandes, Khaloob Kawar, Jelena Kuvač Kraljević, Kakia Petinou, Eleni Theodorou, Tatiana Tumanova, Ioannis Vogandroukas, Carol Westby.

**Methodology:** Marleen F. Westerveld, Rena Lyons, Nickola Wolf Nelson, Sara Ferman, Gail T. Gillon, Carol Westby.

**Project administration:** Marleen F. Westerveld.

**Resources:** Marleen F. Westerveld, Nickola Wolf Nelson, Kai Mei Chen, Mary Claessen, Sara Ferman, Fernanda Dreux M. Fernandes, Gail T. Gillon, Khaloob Kawar, Jelena Kuvač Kraljević, Kakia Petinou, Eleni Theodorou, Tatiana Tumanova, Ioannis Vogandroukas, Carol Westby.

**Supervision:** Marleen F. Westerveld, Gail T. Gillon, Jelena Kuvač Kraljević.

**Validation:** Marleen F. Westerveld, Rena Lyons, Nickola Wolf Nelson, Kai Mei Chen, Mary Claessen, Sara Ferman, Fernanda Dreux M. Fernandes, Khaloob Kawar, Jelena Kuvač Kraljević, Kakia Petinou, Eleni Theodorou, Tatiana Tumanova, Ioannis Vogandroukas, Carol Westby.

**Visualization:** Marleen F. Westerveld, Rena Lyons, Nickola Wolf Nelson, Sara Ferman.

**Writing – original draft:** Marleen F. Westerveld, Rena Lyons, Nickola Wolf Nelson, Carol Westby.

**Writing – review & editing:** Marleen F. Westerveld, Rena Lyons, Nickola Wolf Nelson, Kai Mei Chen, Mary Claessen, Sara Ferman, Fernanda Dreux M. Fernandes, Gail T. Gillon, Khaloob Kawar, Jelena Kuvač Kraljević, Kakia Petinou, Eleni Theodorou, Tatiana Tumanova, Ioannis Vogandroukas, Carol Westby.

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
