## [Decision Letter · Decision Letter 0]

19 May 2022

PONE-D-22-10735Global TALES feasibility study: Personal narratives in 10-year-old children around the worldPLOS ONE

Dear Dr. Westerveld,

Thank you for submitting your manuscript to PLOS ONE. We would like to provisionally accept your manuscript with the understanding that you will make some minor revisions. Therefore, we invite you to submit a revised version of the manuscript that addresses the points raised during the review process.

We look forward to receiving your revised manuscript.

Kind regards,

Sandra Laing Gillam, Ph.D

Academic Editor

PLOS ONE

Journal Requirements:

"MW has a financial relationship with SALT Software. The remaining authors have declared that no competing interests exist."

We note that you received funding from a commercial source: SALT Software

3. One of the noted authors is a group or consortium "Global TALES Consortium". In addition to naming the author group, please list the individual authors and affiliations within this group in the acknowledgments section of your manuscript. Please also indicate clearly a lead author for this group along with a contact email address.

Additional Editor Comments:

Dear Dr. Westerveld,

It was a pleasure reading your manuscript. Two expert reviewers and myself reviewed the paper and agree that it is going to be an important contribution to the literature and potentially clinical practice. I would like to provisionally accept your paper after some very minor revisions. Please address these issues and we will move forward.

We would like to see you include more information for the reader on how you envision the protocol being used in research and clinical practice across culturally and linguistically diverse populations. Along these lines, the reader would profit from more discussion of the pros and cons of greater standardization guidelines for consistent administration of the protocol. One minor edit that should be made is line 100; 3 year olds were not included in the study that you reference, please insert the proper reference. Also, please edit the manuscript so that Tales is stated as "Tales" or "TALES" but not both.

I commend you on an amazing project and look forward to seeing the paper in print.

Sincerely, Dr. Sandra Laing Gillam

Reviewers' comments:

Reviewer's Responses to Questions

**Comments to the Author**

1. Is the manuscript technically sound, and do the data support the conclusions?

Reviewer #1: Yes

Reviewer #2: Yes

2. Has the statistical analysis been performed appropriately and rigorously? 

Reviewer #1: Yes

Reviewer #2: N/A

3. Have the authors made all data underlying the findings in their manuscript fully available?

Reviewer #1: Yes

Reviewer #2: Yes

4. Is the manuscript presented in an intelligible fashion and written in standard English?

Reviewer #1: Yes

Reviewer #2: Yes

5. Review Comments to the Author

Reviewer #1: PLOS ONE

Global TALES feasibility study: Personal narratives in 10-year-old children around the

world

--Manuscript Draft--

Manuscript Number: PONE-D-22-10735. REVIEW

This paper reports on a project to standardize a protocol for eliciting personal narratives from children around the world. It represents a carefully considered effort to collect a type of narrative that all cultures known to date engage in, rather than the rather unconsidered elicitation of fictional narratives, usually from wordless picture books, which are not actually produced by children in any culture and are not anywhere nearly as carefully considered and often have cultural bias built into the pictures and pictured story. (I have known quite proficient child narrators who refused to engage in such production because they preferred inventing their own fictions or having an adult read a set one.).

Results reported in this project included interviewing approximately 20 ten-year-old children from 10 different countries speaking 8 different languages using a fixed set of prompts. They succeeded in coming up with 6 different prompts that got most children producing narratives. While individual variability in production was high (what you want in something that is ultimately intended as a clinical tool), the prompts worked in all countries, as indicated by number of utterances (C-units) and number of words. Topics of narratives were also coded, as were how many narratives were elicited with and without neutral follow-up prompts. Reliability was assessed where needed and found to be acceptable.

The second, linked study was a content analysis of researchers’ experiences in the various countries. This second part of the project was also fascinating. One of the most important findings was that researchers (most speech-language pathologists) were very favorably impressed by the study’s use of personal narratives. They reported to study leaders on all sorts of concerns that were in and of themselves interesting.

Authors have a deep knowledge of prior research on narratives, and a strong command of methodology.

Authors’ goals for examining the feasibility of the TALES project were met, but these are just the beginning of many exciting projects and analyses to come. This is one of the most promising studies this reviewer has encountered in years, and it is to be hoped that many researchers in other countries/languages take them up on their invitation to extend use of TALES. It is a project most appropriate for PLOS audiences.

Minor points:

• Sometimes project is referred to as TALES, sometimes Tales; best to standardize.

• Authors used the word “story” in elicitation, whereas “memory” might have been better, but it is hard to quibble with success.

• Other edits are in appended pdf document.

Reviewer #2: This article describes a study of the feasibility of a protocol for collecting personal narratives from children speaking different languages and living in different cultures. The authors carefully developed a set of prompts to standardize administration as much as possible. They report on general findings of amount of language obtained from children by language and prompt topic. They also categorized the topics of stories obtained and discussed variations in administration, transcription and translation.

Overall, the article fulfills the stated goals of trialling and reporting the initial use of the protocol. The prompts have been revised somewhat to reflect initial findings.

I would like to see a bit more about how the authors envision the uses of the protocol. The analyses in the current paper focused on word and C-unit counts, and that was appropriate for this first pass. It could conceivably be used to compare content of personal stories in terms of cultural content and so forth. The authors allude to clinical uses only indirectly, by mention of making the tool “more attractive for clinicians”. A discussion of how this tool would be useful for cross-linguistic understanding of language development and disorders would provide more context for this protocol.

It would be useful to discuss a bit more the pros and cons of greater standardization of administration. Should administrators be asked to provide a minimum as well as a maximum of prompts? Would that possibly violate some cultural norms? How could that issue be explored further in future studies?

Overall, this work is very interesting and should prove useful for examining the development of personal narratives across languages and cultures.

6. PLOS authors have the option to publish the peer review history of their article (what does this mean?). If published, this will include your full peer review and any attached files.

Reviewer #1: **Yes: **Allyssa Mccabe

Reviewer #2: No

---

## [Author Response · Author response to Decision Letter 0]

4 Jul 2022

These responses are contained in the 'Rebuttal document" - an abbreviated version below:

Author response: 

• We have carefully checked all style requirements and sincerely hope we haven’t missed anything. Some guidelines were ambiguous such as the use of headings or paragraph indentations. 

• Please note that some minor corrections have been made to author names and affiliations.

We have amended the Statement and have included this in the cover letter as requested. Of note, we did not receive funding from SALT Software LLC.

Editor: One of the noted authors is a group or consortium "Global TALES Consortium". In addition to naming the author group, please list the individual authors and affiliations within this group in the acknowledgments section of your manuscript. Please also indicate clearly a lead author for this group along with a contact email address.

Author Response: All named authors are members of the Global TALES Consortium with the main authors named on the author page. Additional members have been named in the acknowledgements, along with their affiliations. There is no lead author for this additional group. We have made this clearer by adding ‘Additional’ to read: Additional members of the Global TALES Consortium. I (Marleen Westerveld am the corresponding author and the leader of the Global TALES Consortium). 

Editor Comments 

We would like to see you include more information for the reader on how you envision the protocol being used in research and clinical practice across culturally and linguistically diverse populations. Along these lines, the reader would profit from more discussion of the pros and cons of greater standardization guidelines for consistent administration of the protocol. 

Author response: We have responded to these issues in our response to the individual reviewers below. In summary, we have re-arranged the final two pages of the manuscript and added two paragraphs to the final section of the paper.

Editor: One minor edit that should be made is line 100; 3 year olds were not included in the study that you reference, please insert the proper reference. Also, please edit the manuscript so that Tales is stated as "Tales" or "TALES" but not both.

Authors: As suggested, we have added a reference to support the inclusion of 3-year-olds. We have now consistently referred to the protocol using capital letters for TALES. 

Reviewer #1: PLOS ONE

Minor points:

• Sometimes project is referred to as TALES, sometimes Tales; best to standardize.

Author Response: this has been corrected – now referred to as TALES throughout. 

• Authors used the word “story” in elicitation, whereas “memory” might have been better, but it is hard to quibble with success.

Author Response: thank you for that suggestion. We agreed to using ‘story’ when developing the protocol so have not changed the wording at this stage of our project. 

• Other edits are in appended pdf document.

Response: thank you. All minor edits have been responded to in the R1 of the paper. See the track changes version of the paper. 

Reviewer #2: 

I would like to see a bit more about how the authors envision the uses of the protocol. The analyses in the current paper focused on word and C-unit counts, and that was appropriate for this first pass. It could conceivably be used to compare content of personal stories in terms of cultural content and so forth. The authors allude to clinical uses only indirectly, by mention of making the tool “more attractive for clinicians”. 

Author response: 

The conclusion sections have been restructured. We have also added the following paragraph to make this clearer:

These findings will also be important for clinicians who work with children from culturally and/or linguistically diverse backgrounds on a daily basis. Eventually, country-specific data, reflecting differences within and across countries and cultures and evaluated for reliability and validity, could result in a practical tool for timely identification of developmental language disorder. From a clinical perspective, having access to an ecologically valid tool will provide important insights into the impact of a child’s communication impairment on their functioning and participation at home, at school, and in the community (see [44]). 

Reviewer:

A discussion of how this tool would be useful for cross-linguistic understanding of language development and disorders would provide more context for this protocol.

It would be useful to discuss a bit more the pros and cons of greater standardization of administration. Should administrators be asked to provide a minimum as well as a maximum of prompts? Would that possibly violate some cultural norms? How could that issue be explored further in future studies?

Author response: Thank you for this suggestion. The final two pages of the manuscript have been restructured. The following text has been added to the conclusion section of the manuscript: 

Despite our promising results, we acknowledge that using a standardized protocol with scripted follow-up prompts may not be readily achievable across all cultures. In an attempt to avoid ‘leading’ the child to create a personal story the examiner takes a fairly passive role when administering the Global TALES protocol. However, in some cultures children may expect stories to be jointly constructed, for example, and further research is needed to evaluate ways of engaging children in story telling in a culturally responsive, yet standardized way to allow for cross-cultural comparisons. Future research should also investigate the effects of specific prompts and/or conversation partners on children’s ability to provide a coherent personal narrative across different cultures, providing much-needed information to enhance cross-linguistic and cross-cultural understanding of language development and disorders. Findings of such research may indicate that, in some cultures, it may not be appropriate for children to discuss feelings and/or talk about certain topics with unfamiliar adults. Regardless, using a standard protocol to collect normative data across a range of languages and cultures could allow researchers to learn more about this universal discourse genre, its development with age or year of schooling, and how culture may impact performance.

---

## [Editor Report · Decision Letter 1]

3 Aug 2022

Global TALES feasibility study: Personal narratives in 10-year-old children around the world

PONE-D-22-10735R1

Dear Dr. Westerveld,

We’re pleased to inform you that your manuscript has been judged scientifically suitable for publication and will be formally accepted for publication once it meets all outstanding technical requirements.

Kind regards,

Sandra Laing Gillam, Ph.D

Academic Editor

PLOS ONE
---

## [Editor Report · Acceptance letter]

5 Aug 2022

PONE-D-22-10735R1 

Global TALES feasibility study: Personal narratives in 10-year-old children around the world 

Dear Dr. Westerveld:

I'm pleased to inform you that your manuscript has been deemed suitable for publication in PLOS ONE. Congratulations! Your manuscript is now with our production department. 

Kind regards, 

on behalf of

Dr. Sandra Laing Gillam 

Academic Editor

PLOS ONE